# OpenReview forum: "Tilt Matching for Scalable Sampling and Fine-Tuning"
_ICML.cc/2026/Conference — ICML 2026 spotlight_

### Official Review · Reviewer_8jhz · 2026-03-07

**Soundness:** 4
**Presentation:** 4
**Significance:** 4
**Originality:** 3
**Overall Recommendation:** 5
**Confidence:** 4

**Summary:**

This paper introduces a method for sampling from tilted distributions. The first part derives an ODE for the evoluation of the tilted drift, which motivates the practical implementation. The second part shows how tilt matching can be used by either ETM which provides a regression objective for obtaining the tilted velocity, although this comes with approximation/discretization errors. Alternatively, the authors proposed ITM which results in a fixed point objective, variance reduction approaches are then discussed.

**Compliance With Llm Reviewing Policy:**

Affirmed.

**Final Justification:**

My concerns were addressed and I decide to keep my positive score

**Key Questions For Authors:**

1. The theoretical analysis around variance reduction via control variates is very interesting. However, I think it would be interesting to see how estimating it impacts results at least in a toy scenario or alternatively what are the main difficulties of integrating the joint optimization in practice.

2. I would like to know how tilt matching compares in terms of computational budget to alternative approaches.

3. In table 1, it is mention that the sample diversity decreases compared to the base model. Could you comment on this and what is the tradeoff between having samples with high reward and maintaining diversity? Is sample diversity affected by the chosen step size as you show how the ESS decreases with h in plot 4c?

**Limitations:**

The authors do not discussed some of the limitations of the methodology, for example the iterative nature of ITM and computational budget or the effect of the step size h.

**Strengths And Weaknesses:**

Strengths:
- Presentation: the paper is very well structured and easy to follow
- Soundness: the methodology is very well supported by theoretical derivations and well connected with practical implementations by proposing different objectives to estimated the velocity field of the tilted path.
-Significance: the problem of sampling from tilted distributions is very relevant in many areas of science. Authors provide empirical evidence of their methodology for lennard jones potential.
- Originality: tilt matching overcomes several shortcoming of previous methods, by avoiding taking gradients of the reward function or differentiating through trajectories.

Weaknesses:
- The iterative nature of ITM can result in a high computational cost.
- Empirical validation is a bit limited. Additionally, it would be nice to include experimental details (e.g. parameter configurations) for the experiments in 4.1.
- Minor: in line 181 should it be '' the conditional covariance between the interpolant's velocity and the reward"?

---

> ### Author Rebuttal · Authors · 2026-03-30
>
> We thank the reviewer for their kind review and for finding the control variate part of our paper interesting.
>
> ### Answer to Question 1:
>
> We ran a simple toy experiment to test whether learning the control variate helps in practice. The base terminal distribution is a bimodal Gaussian mixture $p_1 = 0.5 \cdot \mathcal{N}(-1, 1) + 0.5 \cdot \mathcal{N}(1, 1)$, and we consider a simple reward tilt $e^{r(x)} = 4 \text{ if } x \leq 0$, and $e^{r(x)} = 5 \text{ if } x > 0$. We use ITM with a step size $h = 1$.
>
> We compare three variants: no control variate $(c=0)$, a fixed control variate $(c=1)$, and a learned/trainable control variate. We monitor the variance of the training gradient throughout optimization, as well as the Wasserstein $W_2$ distance between model samples and the exact reweighted target. The results are shown below:
>
> | Method | **Step 1** |  | **Step 5,000** |  | **Step 25,000** |  |
> |:---|:---:|:---:|:---:|:---:|:---:|:---:|
> |  | Grad Var | $W_2$ | Grad Var | $W_2$ | Grad Var | $W_2$ |
> | $c=0$ | 2.409 | 0.589 | 2.127 | 0.091 | 2.235 | 0.089 |
> | $c=1$ | 1.905 | 0.589 | 1.599 | 0.083 | 1.699 | 0.072 |
> | learnable $c$ | 1.905 | 0.589 | 0.147 | 0.057 | 0.046 | 0.051 |
>
> In this setting, the learned control variate substantially reduces the variance of the gradient throughout training, while also improving convergence to the exact reweighted target as measured by the $W_2$ distance. This supports the claim that joint learning of the control variate can provide a practical optimization benefit, even in a simple toy setting.
>
> For the large-scale experiments, integrating a trainable control variate is also straightforward in practice. We did not include it mainly to keep the presentation focused on the core method and because, for the small annealing steps $h$ used in our iterative setting, the optimal control variate is already close to $1$, so the additional gains are expected to be smaller.
>
> ### Answer to Question 2:
>
> A direct computational comparison is somewhat nuanced because ITM and trajectory-based methods such as Adjoint Matching (AM) incur different kinds of cost. Per gradient step, AM requires differentiating through the reward and evaluating the network along a trajectory, leading to substantially more network evaluations than a single ITM update. In contrast, ITM avoids reward gradients and trajectory backpropagation, but typically requires more gradient updates overall. In addition, the practical runtime of both methods depends strongly on implementation details, including how trajectories or samples are generated, reused in buffers, and refreshed during training. In our implementation, which builds on the AM codebase, we chose the training settings so that **ITM's runtime matched that of AM**. We also emphasize that in settings where the reward comes from an external simulator or expensive system, reward differentiation may be unavailable or prohibitively costly, making AM impractical while ITM remains feasible since it only requires reward evaluations.
>
> ### Answer to Question 3:
>
> We agree that there is a tradeoff between reward and diversity: increasing the tilt strength $\lambda$ concentrates the target distribution $p(x)\exp(\lambda r(x))$ on higher-reward regions, which can reduce diversity because probability mass shifts toward a smaller region of the sample space. In our setting, however, the goal is to sample accurately from this specified tilted distribution. Thus, the appropriate way to adjust the reward/diversity tradeoff is to change the target itself, e.g. via $\lambda$, rather than to deviate from the target during optimization. This differs from reinforcement learning-style fine-tuning, where one often seeks high reward samples without a uniquely specified target distribution. In such settings, the objective and algorithm must be modified accordingly to explicitly encode the desired reward/diversity tradeoff, for example through additional regularization terms.
>
> Regarding Figure 4c, lower ESS indicates that the sampler is less faithful to the intended target as the step size $h$ increases. It is therefore unclear whether a change in ESS would systematically improve or harm diversity. For this reason, we would not want to control diversity by varying the step size; instead, diversity should be adjusted by altering the target distribution itself. We therefore think the reduced diversity in Table 1 is likely a consequence of more successfully moving toward the reward-tilted target distribution.
>
> ### Other
> Thank you for catching this typo. Yes, this should read "the conditional covariance between the interpolant’s velocity and the reward" and we will correct the wording in the revision.

---

> > ### Author Rebuttal · Reviewer_8jhz · 2026-04-02
> >
> > Thanks for the reply! I think the discussion on the ESS is very interesting. My concerns have been acknowledge

---

> > > ### Author Response · Authors · 2026-04-06
> > >
> > > We are pleased to hear that our rebuttal has addressed your questions, and we would like to thank the reviewer for their positive and helpful comments during the review process.
> > >
> > > Thanks,
> > >
> > > The Authors

---

### Official Review · Reviewer_FsXd · 2026-03-12

**Soundness:** 2
**Presentation:** 3
**Significance:** 2
**Originality:** 3
**Overall Recommendation:** 4
**Confidence:** 4

**Summary:**

The paper proposes Tilt Matching (TM), a scalable framework to adapt flow/diffusion generative transports to reward-tilted target distribution $\rho_{1,a} \propto \rho_1 e^{ar}$. The core idea is to characterize the tilted transport field and use this to define practical training objectives. The paper also draws connections to Doob's $h$-transform and SOC. Experiments are conducted on Lennard-Jones sampling and SDv1.5 fine-tuning.

**Compliance With Llm Reviewing Policy:**

Affirmed.

**Final Justification:**

Most of my concerns are addressed.

**Key Questions For Authors:**

Please see weakness

**Limitations:**

Yes.

**Strengths And Weaknesses:**

**Strengths**

1. The paper is well motivated and develops a theoretical connection between reward tilting, transport dynamics, and SOC. The covariance ODE is interesting and nontrivial.
2. The proposed objectives are conceptually good and practically appealing since they avoid reward gradients and backpropagation through trajectories.
3. The experimental results on Lennard-Jones sampling outperform other baselines.

**Weakness**

1. The paper argues that ITM addresses the first-order approximation issue in ETM. However, ITM still relies on the approximation $\hat b_{t, a}(\cdot)\approx b_{t, a+h}(\cdot)$, so it is not clear whether this is actually better than the first-order approximation of ETM in practice. A direct comparison between ETM and ITM, including training stability, would strengthen the paper.

2. In Tab. 1, Adjoint Matching (AM) with $\lambda = 10^2$ outperforms ITM in both target reward and held-out rewards. Since the paper mentions that further gains may be possible with hyperparameter sweeps, it would be important to also report ITM results with $\lambda = 10^2$ for a fair comparison under the same target distribution.

3. I have some concerns on the scalability of the framework. Although the paper emphasizes that TM avoids reward gradients and trajectory backpropagation, it does not provide a direct comparison of runtime or training stability against trajectory-based methods such as Adjoint Matching. In addition, TM requires repeated offline fine-tuning along the annealing schedule $\\{a_k\\}_{k=0}^K$, which may introduce nontrivial overhead and sensitivity to imperfect convergence at intermediate stages. Thus, without such evidence, the practical efficiency advantage remains unclear.

---

> ### Author Rebuttal · Authors · 2026-03-30
>
> We thank the reviewer for their kind review and for finding our theoretical development interesting.
>
> ### Answer to Question 1:
>
> First, we emphasize that the ITM objective **is not** based on the approximation $\hat b_{t, a}(\cdot)\approx b_{t, a+h}(\cdot)$. ITM is defined through the fixed-point condition in Eq. (18), whose unique solution is the exact target drift $b_{t,a+h}$. So the approximation issue present in ETM is removed in the ITM objective.
>
> We agree with the reviewer that the practical question is whether this manifests in improved empirical performance, as we show below. For the Stable Diffusion experiments, we provide the following comparison:
>
> | Method             | ImageReward (↑)        |
> |--------------------|------------------------|
> | SD 1.5 (Base)      | 0.1873 ± 0.0762        |
> | AM (λ=1)            |       0.2170 ±  0.0755 |
> | ETM (λ=1)          | 0.3799 ± 0.0744        |
> | ITM (λ=1)          | 0.4465 ± 0.0709        |
>
> Under identical training settings, these results demonstrate a clear improvement of ITM over ETM on Stable Diffusion. For the LJ experiments, ETM does not converge under the same fixed-step setting used for ITM, yielding a final ESS below 0.01, and the corresponding energy plots exhibit clear mode collapse.
>
> We thank the reviewer for the suggestion and will include these additional comparisons in the paper to provide stronger evidence of the practical advantages of ITM over ETM.
>
> ### Answer to Question 2:
>
> We agree that an ablation over the tilt coefficient would be informative. That said, we do not believe that setting ITM to $\lambda=10^2$ solely to match AM would be a like-for-like comparison. In particular, it is unlikely that AM at $\lambda=10^2$ should be interpreted literally as targeting the distribution $p(x)e^{100r(x)}$. Such a value corresponds to an extremely aggressive tilt and would likely lead to severe reward hacking, especially since $\lambda=1$ already produces noticeable changes. Instead, the large multiplier appears to serve mainly to amplify a reward-gradient signal that is otherwise washed out by noise. In this sense, $\lambda=10^2$ functions less as a literal target specification and more as an optimization device for AM, making the comparison harder to interpret. By contrast, ITM does not rely on reward gradients and uses only scalar reward evaluations, so the role of $\lambda$ is fundamentally different. Comparing only at $\lambda=10^2$ would therefore not be very informative; a fairer comparison would require a broader grid search over $\lambda$ for ITM as well, and we will include it in the camera ready version.
>
> ### Answer to Question 3:
>
> A direct computational comparison is somewhat nuanced because ITM and trajectory-based methods such as Adjoint Matching (AM) incur different kinds of cost. Per gradient step, AM requires differentiating through the reward and evaluating the network along a trajectory, leading to substantially more network evaluations than a single ITM update. In contrast, ITM avoids reward gradients and trajectory backpropagation, but typically requires more gradient updates overall. In addition, the practical runtime of both methods depends strongly on implementation details, including how trajectories or samples are generated, reused in buffers, and refreshed during training. In our implementation, which builds on the AM codebase, we chose the training settings so that **ITM's runtime matched that of AM**. We also emphasize that in settings where the reward comes from an external simulator or expensive system, reward differentiation may be unavailable or prohibitively costly, making AM impractical while ITM remains feasible since it only requires reward evaluations.
>
> Regarding the repeated offline fine-tuning along the annealing schedule, we emphasize that the ITM objective itself does not require an annealing schedule (in contrast to ETM, which is derived as a stepwise update in $a$). For ITM, we include annealing as an implementation choice that improves training behaviour. All of **our reported experiments already use an annealing schedule**, so the empirical results show that ITM is effective in this regime.
>
>
> We hope that our responses address the reviewer’s main concerns, and we kindly ask them to consider increasing their score if they agree.

---

> > ### Author Rebuttal · Reviewer_FsXd · 2026-04-03
> >
> > Thank you for your detailed response.
> > Most of my concerns are fully addressed.

---

> > > ### Author Response · Authors · 2026-04-06
> > >
> > > We would like to thank the reviewer for their response, and for their valuable comments during the review process. We appreciate the reviewer raising their score.
> > >
> > > Thanks,
> > >
> > > The Authors

---

### Official Review · Reviewer_Wsmk · 2026-03-12

**Soundness:** 2
**Presentation:** 2
**Significance:** 2
**Originality:** 3
**Overall Recommendation:** 4
**Confidence:** 4

**Summary:**

This paper proposes Tilt Matching, a scalable algorithm for sampling from reward-tilted distributions and for fine-tuning generative models such as diffusion or flow-based models. The method is built on the stochastic interpolant / flow matching framework. Instead of directly optimizing the velocity field for the reward-tilted distribution using reinforcement learning or trajectory-level optimization, the paper derives a dynamical relationship between the base flow velocity and the velocity corresponding to the tilted distribution. The key insight is that the updated velocity field can be expressed as an expansion involving joint cumulants between the interpolant velocity and the reward function. To first order, this reduces to a covariance between the velocity and the reward, yielding a simple estimator for updating the velocity field.

**Compliance With Llm Reviewing Policy:**

Affirmed.

**Final Justification:**

My concerns have been adequately addressed.

**Key Questions For Authors:**

See weakness.

**Limitations:**

See questions.

**Strengths And Weaknesses:**

**Strengths:**

1. The paper introduces a clean theoretical framework that connects reward tilting with flow matching dynamics. By deriving a relation between the base velocity and the reward-tilted velocity through cumulant expansions, the method provides an interpretable perspective on how reward signals modify transport dynamics.

2. A key advantage of Tilt Matching is that the optimization objective has lower variance than standard flow matching objectives. The proposed estimator, which reduces to a covariance between velocity and reward, is simple and efficient.

**Weakness:**

1. It would be helpful if the paper could explicitly present the sampling and fine-tuning procedures based on the proposed ITM framework. While the theoretical formulation is described, the paper does not clearly outline the concrete algorithms used for sampling and model fine-tuning.

2. The paper claims that the proposed method requires neither reward gradients nor backpropagation through the trajectories of the flow or diffusion process. It would be helpful if the authors could provide a quantitative comparison of the fine-tuning overhead when using ITM relative to other reward-based fine-tuning methods, such as AM.

3. The paper proposes improving Tilt Matching by transitioning from ETM to ITM to reduce discretization errors. It would strengthen the paper if the authors could provide more empirical evidence demonstrating that the performance gains from ETM to ITM. Besides, the detailed experimental setting are missing in Sec. 4.

---

> ### Author Rebuttal · Authors · 2026-03-30
>
> We thank the reviewer for their kind review and for finding our theoretical framework interesting.
>
> ### Answer to Question 1:
>
> Thank you, we agree that the concrete procedures should be presented more explicitly, and in the revised version we will move the full ITM algorithm into the main body. Currently, it is given as Algorithm 2 in the appendix. At a high level, the core algorithm is the same for both sampling and model fine-tuning; the main difference is how the reward is defined in each application. We will make these implementation details explicit in the revision.
>
> ### Answer to Question 2:
>
> A direct computational comparison is somewhat nuanced because ITM and trajectory-based methods such as Adjoint Matching (AM) incur different kinds of cost. Per gradient step, AM requires differentiating through the reward and evaluating the network along a trajectory, leading to substantially more network evaluations than a single ITM update. In contrast, ITM avoids reward gradients and trajectory backpropagation, but typically requires more gradient updates overall. In addition, the practical runtime of both methods depends strongly on implementation details, including how trajectories or samples are generated, reused in buffers, and refreshed during training. In our implementation, which builds on the AM codebase, we chose the training settings so that **ITM's runtime matched that of AM**. We also emphasize that in settings where the reward comes from an external simulator or expensive system, reward differentiation may be unavailable or prohibitively costly, making AM impractical while ITM remains feasible since it only requires reward evaluations.
>
>
> ### Answer to Question 3:
>
> We agree with the reviewer that demonstrating the improved empirical performance of ITM over ETM would help strengthen the paper. For the Stable Diffusion experiments, we provide the following comparison:
>
> | Method             | ImageReward (↑)        |
> |--------------------|------------------------|
> | SD 1.5 (Base)      | 0.1873 ± 0.0762        |
> | AM (λ=1)            |       0.2170 ±  0.0755 |
> | ETM (λ=1)          | 0.3799 ± 0.0744        |
> | ITM (λ=1)          | 0.4465 ± 0.0709        |
>
> Under identical training settings, these results demonstrate a clear improvement of ITM over ETM on Stable Diffusion. For the LJ experiments, ETM does not converge under the same fixed-step setting used for ITM, yielding a final ESS below 0.01, and the corresponding energy plots exhibit clear mode collapse.
>
> We thank the reviewer for the suggestion and will include these additional comparisons in the paper to provide stronger evidence of the practical advantages of ITM over ETM. We will also add detailed experimental settings to Section 4.
>
> We hope that our responses address the reviewer’s main concerns, and we kindly ask them to consider increasing their score if they agree.

---

> > ### Author Rebuttal · Reviewer_Wsmk · 2026-04-03
> >
> > My concerns have been adequately addressed.

---

> > > ### Author Response · Authors · 2026-04-06
> > >
> > > We would like to thank the reviewer for their valuable comments during the review process which have helped to improve our paper. We are glad that our rebuttal was helpful, and we appreciate the reviewer raising their score.
> > >
> > > Best,
> > >
> > > The Authors

---

### Official Review · Reviewer_youh · 2026-03-13

**Soundness:** 3
**Presentation:** 4
**Significance:** 3
**Originality:** 4
**Overall Recommendation:** 5
**Confidence:** 3

**Summary:**

The paper introduces Tilt Matching, a new algorithm based on stochastic interpolants for sampling from unnormalized distributions and fine-tuning generative models based on a reward. The authors derive a differential equation—the Covariance ODE—that describes the evolution of the flow matching velocity field under this exponential tilt, showing the infinitesimal change is driven by the conditional covariance between the interpolant dynamics and the reward. They propose two regression-based objectives: Explicit Tilt Matching (ETM), which suffers from discretization bias, and Implicit Tilt Matching (ITM), a fixed-point objective that eliminates this bias. ITM bypasses the need for backpropagating through trajectories or computing reward gradients. The authors establish theoretical connections between their drift and Doob's h-transform. Empirically, ITM is shown to achieve state-of-the-art Effective Sample Size (ESS) on Lennard-Jones molecular sampling and competitive performance when fine-tuning Stable Diffusion 1.5.

**Compliance With Llm Reviewing Policy:**

Affirmed.

**Final Justification:**

Thank you for the rebuttal. The answers to Q1 and Q4 are convincing and strengthen my confidence in the paper. Q2 is reasonable, though not fully conclusive empirically, and Q3 remains an open suggestion for strengthening the final version. Overall, the rebuttal maintains my positive assessment.

**Key Questions For Authors:**

- How does the wall-clock training time of ITM compare to ETM in practice, and did you empirically observe ETM's discretization error negatively impacting sample quality?

- Do you have any reasoning about the results on fine tuning stable diffusion get less diversity than alternative methods?

- It would be great to see the boltzman sampling capability tested on a harder task, like alanine dipeptide or tripeptide.

- Would it be possible to implement this tilting over a pre-trained flow matching model, instead of a stochastic interpolant?

**Limitations:**

Yes

**Strengths And Weaknesses:**

**Soundness**: The theoretical foundation is rigorous. The proofs are through and well laid out. The connection drawn between Tilt Matching and stochastic optimal control does a good job at situating the work within the broader literature. The results on the Stable Diffusion 1.5 benchmark are very solid, though it would be worth exploring why diversity is worse than for alternative methods. The results on Boltzmann distributions are also quite impressive, though it is true that LJ, while difficult, is still a toy problem. It would be interesting to see benchmarks on more difficult targets like alanine dipeptide or tripeptide.

**Presentation**: The paper is mathematically dense but clearly structured. The progression from the core problem to the Covariance ODE, then to the explicit discretization and finally to the implicit formulation; has a nice flow. Figure 2 provides an excellent schematic intuition.

**Significance**: Both fine-tuning generative models, and sampling from unnormalized probability distributions are difficult problems and the focus of a lot of work. The contribution is novel, and shows a strong performance.

**Originality**: While tilting distributions is a classic concept, deriving an exact relationship between a stochastic interpolant's velocity field and its reward-tilted counterpart via conditional covariance is highly original. The formulation of the loss to match infinite cumulant expansions without trajectory differentiation is a novel contribution to the generative modeling literature

---

> ### Author Rebuttal · Authors · 2026-03-30
>
> We thank the reviewer for their kind review and for finding our theoretical development interesting.
>
> ### Answer to Question 1:
>
> At a fixed annealing step size $h$, ETM and ITM have essentially the same training cost. Both use samples from the current model, the same interpolant quantities, the same reward evaluations, and the same network forward/backward passes. The only difference is the precise expression in the loss used to regress the next velocity field. That said, ETM may need smaller $h$ (and therefore more annealing stages) to reduce discretization bias and achieve the same performance as ITM.
>
> Empirically, ITM consistently outperforms ETM when using the same fixed step size $h$. For the Stable Diffusion experiments, we observe the following:
>
> | Method             | ImageReward (↑)        |
> |--------------------|------------------------|
> | SD 1.5 (Base)      | 0.1873 ± 0.0762        |
> | AM (λ=1)            |       0.2170 ±  0.0755 |
> | ETM (λ=1)          | 0.3799 ± 0.0744        |
> | ITM (λ=1)          | 0.4465 ± 0.0709        |
>
> Since ETM and ITM use the same training settings, the performance gap suggests that ETM’s discretization bias is not merely theoretical, but affects sample quality in practice.
>
> ### Answer to Question 2:
>
> We agree that there is a reward/diversity tradeoff here. Targeting the reward-tilted distribution $p(x)\exp(r(x))$ naturally shifts probability mass toward higher-reward regions, which can reduce diversity relative to the base distribution. We emphasize that our goal is to sample exactly from the reward-tilted target distribution, so the correct notion of diversity is the diversity of that target distribution itself. If the target is more concentrated than the base distribution, then a decrease in the diversity metric is consistent with better target matching rather than worse behaviour.
>
> In our Stable Diffusion results, ITM achieves a higher ImageReward than AM, while CLIPScore and HPSv2 remain similar. At the same time, DreamSim decreases. We therefore think the lower diversity is likely a consequence of more successfully moving toward the reward-tilted target distribution, rather than simply an optimization artifact. That said, it is difficult to verify this claim for the text-to-image experiments as the true target density is not directly accessible. By contrast, in the Lennard-Jones setting the target Boltzmann distribution is explicitly specified, which lets us confirm more directly that the method tracks the intended tilted target.
>
>
> ### Answer to Question 3:
>
> Given the rebuttal timeline, we unfortunately do not have the computational resources to complete this experiment that quickly, but we would be happy to include it in the camera ready version if accepted.
>
> ### Answer to Question 4:
>
> Yes, that is definitely possible! In fact, in our Stable Diffusion experiment, we reparameterize the model in flow matching form. In particular, the stochastic interpolant framework recovers the flow matching one by choosing the coefficients $\alpha(t)=1-t$ and $\beta(t)=t$, up to reversing the time convention so that $t=0$ corresponds to noise and $t=1$ to data.

---

> > ### Author Rebuttal · Reviewer_youh · 2026-04-08
> >
> > The authors have addressed enough of my questions to maintain my original score

---

### Decision · Program_Chairs · 2026-04-30

**Decision:**

Accept (spotlight)

**Comment:**

This nicely written paper proposes Tilt Matching, an approach to tilt a pretrained generative model towards fulfilling a specified reward. All reviewers recommend acceptance of the paper, confirming the technical quality, validity, and interest. While they were concerned about the runtime (reviewer FsXd, 8jhz), authors assured that it was comparable with competing approaches. Concerns on the reduction of diversity (reviewer FsXd, 8jhz) were convincingly explained by the authors. I therefore strongly suggest accepting the paper to the program. Authors, please include the promised experiments on less toyish molecule data such as alanine dipeptide or tripeptide (reviewer youh). I also strongly encourage the authors to include a more hands-on formulation of Algorithms 1 and 2 to better communicate the structure of the training algorithm.